# Trends in Antibiotic Resistance of Nosocomial and Community-Acquired Infections in Italy

**DOI:** 10.3390/antibiotics12040651

**Published:** 2023-03-24

**Authors:** Paola Cerini, Francesca Rita Meduri, Flaminia Tomassetti, Isabella Polidori, Marta Brugneti, Eleonora Nicolai, Sergio Bernardini, Massimo Pieri, Francesco Broccolo

**Affiliations:** 1Cerba Healthcare Italia srl, Viale Roma 190/A, Guidonia Montecelio, 00012 Rome, Italy; 2Microbiology and Virology Laboratory, Department of Molecular Medicine, University of Rome “La Sapienza”, 00185 Rome, Italy; 3Biochemistry Laboratory, Department of Experimental Medicine, University of Rome “Tor Vergata”, 00133 Rome, Italy; 4Microbiology and Virology Laboratory, Department of Experimental Medicine, University of Rome “Tor Vergata”, 00133 Rome, Italy; 5Department of Laboratory Medicine, “Tor Vergata” University Hospital, Viale Oxford 81, 00133 Rome, Italy; 6Department of Biological and Environmental Sciences and Technologies (DiSTeBA), University of Salento, 73100 Lecce, Italy

**Keywords:** antibiotic resistance, nosocomial infections, community infections

## Abstract

The World Health Organization has recently identified three categories of pathogens, namely: critical, high, and medium priority, according to the need for new antibiotics. Critical priority pathogens include carbapenem-resistant microorganism (CPO) such as *A. baumannii* and *P. aeruginosa*, *K. pneumoniae*, and *Enterobacter* spp., whereas vancomycin-resistant *E. faecium *(VRE)*,* methicillin and vancomycin-resistant *S. aureus* (MRSA) are in the high priority list. We compared the trend of antimicrobial resistants (AMRs) in clinical isolates, divided by year and bacteria *spp.*, of samples obtained from nosocomial and community patients. Patient records were collected, including age, sex, site of infection, isolated organisms, and drug susceptibility patterns. From 2019 to 2022, a total of 113,635 bacterial isolates were tested, of which 11,901 resulted in antimicrobial resistants. An increase in the prevalence of several antibiotics resistant bacteria was observed. Specifically, the percentage of CPO cases increased from 2.62% to 4.56%, the percentage of MRSA increased from 1.84% to 2.81%, and the percentage of VRE increased from 0.58% to 2.21%. AMRs trend resulted in increases in CPO and MRSA for both community and nosocomial. Our work aims to highlight the necessity of preventive and control measures to be adopted in order to reduce the spread of multidrug-resistant pathogens.

## 1. Introduction

Antimicrobial resistance is the ability of microorganisms to survive under the presence of antimicrobial substances at concentrations usually sufficient to inhibit or kill them. This problem concerns clinicians and highlights a serious threat to public health, making it a global priority for healthcare [1]. The presently available antibiotics are less and less effective on microorganisms, and it is essential to develop new strategies to manage them. Since the 1950s, this phenomenon has increased and is still increasing in recent years, affecting countries, and increasing morbidity and mortality. In particular, Italy ranks first in Europe for the high prevalence of resistant bacterial strains and mortality attributable to antibiotic resistance [2]. AMR is a rising global epidemic that is typically attributed to “selective pressure” brought on by excessive, or inappropriate use of antibiotics in people and animals [3].

Different types of bacterial resistance can be distinguished, including natural and acquired resistance, and each one can act with different mechanisms [4]. The resistance mechanisms developed by bacteria are essentially four: specific modification of the target, leading to the loss or decrease of the drug’s affinity for its target; enzymatic inactivation of the antibiotic with modification of its active molecular part; decreased permeability, which results in a decrease in the number or diameter of the porins in the outer membrane of the Gram-negatives, making it impossible for the drug to enter or be absorbed at a lower rate; and expulsion of the antibiotic from the cell through energy-dependent efflux pumps. For example, resistance can be achieved through the inactivation of antibiotics mediated by the expression of enzymes such as β-lactamases [4]. Often, a microorganism can develop different resistance mechanisms [5,6]. 

The WHO shared a list of “priority status” pathogens, defined as ESKAPE (*Enterococcus faecium*, *Staphylococcus aureus*, *Klebsiella pneumoniae*, *Acinetobacter baumannii*, *Pseudomonas aeruginosa*, and *Enterobacter* species) pathogens to focus and guide the development of new antibiotics [7,8]. Globally, rates of resistance are constantly increasing, of which 40–70% are carbapenems-resistant infections [9]. Carbapenemases, also called broad-spectrum β-lactamases, can induce resistance to imipenem and meropenem of *Pseudomonas aeruginosa* and other Gram-negative rods. The carbapenem-resistant organism (CPO) mainly includes carbapenem-resistant Enterobacteriaceae (CRE), carbapenem-resistant *A. baumannii* (CRAB), and carbapenem-resistant *P. aeruginosa* (CRPA). Likewise, Vancomycin-resistant enterococci (VREs) have emerged in recent years as epidemiologically relevant pathogens, posing challenges both for case management for multidrug resistance and from a public health perspective due to the potential risk of transferring vancomycin resistance genes to other Gram-positive organisms. However, before the CPO and VRE, *S. aureus* is the most well-known example of an AMR, defining the Methicillin-resistant *S. aureus* (MRSA) [10].

Nonetheless, in recent years, AMR has been leaking from hospital units [11,12,13,14,15,16], increasingly escaping from the hospital environment among the local population, which led the authorities such as the CDC (Center for Disease Prevention and Control) and WHO to activate numerous surveillance systems, based on the data collection at a local or national level [17]. The epidemiology of AMR globally appears to be very diverse, with significant differences between countries; in some cases, large-scale epidemics have involved numerous hospitals in the same region. In other contexts, the presence of these microorganisms has become endemic, while there are countries where the phenomenon is emerging. However, the epidemiological situation in other Italian regions is not known, nor is the level of diffusion of EPCs in non-hospital settings. Our research could provide a preliminary overview. This study focuses on analyses of CRE, CRAB, CRPA, VRE, and MRSA isolated in patients from hospital environments, trying to compare the results to antibiotic resistance isolated in Italian patients from a non-nosocomial ambient from 2019 to 2022. Our focus was on the “ESKAPE” pathogens with carbapenem resistance, Vancomycin-resistant enterococci (VREs), and methicillin-resistant *S. aureus* (MRSA).

## 2. Results

From 2019 to 2022, a total of 113,635 bacterial isolates were executed, of which 11,901 were positive for AMR (CPO, MRSA, and VRE). The positive rate out of the total isolated samples was 10.47% among the Italian population. Table 1 reports the positive resistance percentages divided by antimicrobial substances over the four years. A slight increase was observed for each resistance: from 2.62% (564 out of 17,092) in 2019 to 4.56% (1570 out of 26,734) in 2022 for CPO; from 1.84% (397 out of 2069) in 2019 to 2.81% (966 out of 4062) in 2022 for MRSA; from 0.58% (124 out of 2385) in 2019 to 2.21% (761 out of 3615) in 2022 for VRE.

The AMRs growth over the four-year period is clearly revealed in the graph in Figure 1, which shows the percentages of positivity for each resistance (CPO, MRSA, and VRE) divided by year.

Figure 2 describs the AMRs trend during the study period divided by bacterial spp. An increase in the positive rate has been registered for all the species except the *Acinetobacter baumanii*. The sharp increase of *E. faecium* from 37% (55 out of 149) to 96% (602 out of 625) can be clearly seen, as well as the increase of *S. aureus* from 33% (397 out of 1206) to 57% (966 out of 1698). Less evident from the graph, but equally consistent, is the increase observed for *E. coli* (0.39% in 2019, 48 out of 12,163; 0.26% in 2020, 37 out of 14,178; 0.23% in 2021, 41 out of 17,519; 1% in 2022, 60 out of 5276); for *P. auroginosa* from 5% (82 out of 1565) to 11% (296 out of 2654); and for *E. faecalis* (3% in 2019, 69 out of 2236; 4% in 2020, 99 out of 2598; 6% in 2021, 186 out of 2950; 5% in 2022, 159 out of 2989). A wavering trend, though growing, was reported in *Klebsiella pneumonieae* (7% in 2019, 228 out of 3052; 11% in 2020, 421 out of 3681; 12% in 2021, 536 out of 4450; 5,31% in 2022, 974 out of 18,359), while in *A. baumannii* isolates it was noted a moderate decrease from 67% (206 up to 308) to 53% (240 up to 454). 

In Table 2, the data is divided into samples from nosocomial patients (N) and samples from community patients (C). The detection rate of CPO in nosocomial and community patients is compared from 2019 to 2022. In 2019 and 2021, the detection rates of CPO were slightly higher in nosocomial patients than in community patients (2019: N 39%, C 34%, χ^2^ 3.94, *p* < 0.05, 2021: N 35%, C 28%, χ^2^ 18.46, *p* < 0.001), while in 2022 (N 28%, C 42%, χ^2^ 95.44, *p* < 0.001), the trend was inverted. The detection rate of MRSA showed an opposite trend over the year, with a higher detection rate in community patients compared to nosocomial patients (N) (2019: N 53%, C 59%, χ^2^ 6.79, *p* < 0.05; 2022: N 56%, C 42%, χ^2^ 78.55, *p* < 0.001). Lastly, the trend in the detection rate of VRE was quite stable over the years, with no significant differences observed except for 2021, where the rate of samples from community patients was higher than the rate of samples from nosocomial patients (N 11%, C 16%, χ^2^ 20.30, *p* < 0.001). Analyzing the resistances for a single species revealed that in 2019, no significant difference was observed (*p*: ns). In 2020, carbapenem-resistance *E. coli* was mostly detected in samples from nosocomial patients (N 7%, C 0%, χ^2^ 28.09, *p* < 0.001) as well as MRSA (N 44%, C 36%, χ^2^ 7.70, *p* < 0.05), while carbapenem-resistance *K. pneumoniae* was higher in community patients than nosocomial patients (N 43%, C 52%, χ^2^ 7.11, *p* < 0.05). In 2021, the data confirmed the trend of MRSA (N 44%, C 35%, χ^2^ 4.31, *p* < 0.05), and significant differences were observed in vancomycin-resistance *E. faecalis* (N 55%, C 34%, χ^2^ 16.31, *p* < 0.001) and vancomycin-resistant *E. faecium* (N 45%, C 66%, χ^2^ 4.87, *p* < 0,05). In 2022, higher detection rates for samples from nosocomial patients than community patients were observed in carbapenem-resistant *A. baumannii* (N 24%, C 4%, χ^2^ 121.6, *p* < 0.001), and carbapenem-resistant *E. coli* (N 6%, C 1%, χ^2^ 19.18, *p* < 0.001). On the other hand, an increase in the detection rate of samples from community patients over the nosocomial patients was observed in carbapenem-resistant *K. pneumoniae* (N 51%, C 76%, χ^2^ 102.01, *p* < 0.001), and vancomycin resistant *E. faecalis* (N 17%, C 29%, χ^2^ 10.98, *p* < 0.001).

Moreover, Figure 3 reports graphically the data of resistance detection rate from samples of nosocomial patients vs. community patients.

## 3. Discussion

In recent years, the speed of diffusion of AMR, responsible for both nosocomial and community-based infections, has reached alarming levels [18]. Enterobacteriaceae, especially *E. coli* and *K. pneumoniae*, can become pathogens through the acquisition of resistance phenotypes such as the production of carbapenemases [19]. 

The attention of the scientific community has been directed toward the production of carbapenemases in Enterobacteriaceae [20], however, recently, the interest is twisting toward Vancomycin resistance in Enterococcus, too [21,22]. The clinical relevance of VRE is due to their ability to cause a wide spectrum of infections, most commonly urinary tract infection (UTI), intraabdominal infection, bacteremia, or endocarditis, and rarely can cause meningitis, osteomyelitis, and pneumonia [23]. The spread of VRE has become an emergency of public interest, especially since the infection and colonization by these multi-resistant pathogens have been associated with higher mortality in hospital settings. In 2019, the WHO estimated 133,000 deaths attributable to AMR in Europe [24], implementing the surveillance against antimicrobial resistance with the National Action Plans (NAP) against Antimicrobial Resistance. Compared to previous years, the VRE rate is certainly increased thanks to the surveillance carried out in hospitals. The significant overall increase in cases seen after 2020 could be due to the rise of access in the hospital for SARS-CoV-2 infection recovery, which has led clinicians to prioritize and monitor the surveillance in all the reports. The results obtained in this study emphasized the importance of introducing active screening in hospitals and even in the community to identify vehicles of AMR infection to prevent further transmission. Furthermore, active screening as part of a broader infection prevention program can contain colonization rates. The percentages of resistance to the main classes of antibiotics for the pathogens under surveillance, compared with the national trend, remain high and/or rising over the years [25]. In particular, the incidence of nosocomial VRE infections is increased progressively in Italy, as well as in Europe [8,26], and analyzing, among the positive samples, the isolation data of *K. pnaumoniae*, *S. aureus,* and *E. faecium*, our results seem to confirm this trend. It can be noted, as shown in Table 2, a constant increase over the years starting from 2019 up to 2022, in parallel to NAP against Antimicrobial Resistance [27].

It cannot be excluded that the increased incidence of antimicrobial resistance could be attributed to the excess use of antimicrobial agents during the coronavirus disease 2019 (COVID-19) pandemic¸ which can be seen in Figure 2 by the increase in resistance trend for 2021 and 2022 in *E. coli* (0.23% and 1%), *P. aeruginosa* (9% and 11%), *S. aureus* (47% and 57%), and *E. faecium* (60% and 96%). Patients with COVID-19 have been vulnerable to other secondary infections owing to multiple comorbidities with severe COVID-19, prolonged hospitalization, and severe acute respiratory syndrome coronavirus 2 (SARS-CoV-2)-associated immune dysfunction. These hospital patients have often acquired secondary bacterial infections [28], which can increase the odds of developing AMR; however, even in community patients, resistance growth has been observed in the last two years, especially for *K. pneumoniae* (2021: nosocomial, N, 56% vs. community, C 51%; 2022: N 51% vs. C 76%) and *E. faecium* (2021: N 45% vs. C 66%; 2022: N 83% vs. C 71%), possibly being more sensitive to other infections for the same excess in the use of antibiotics. Moreover, the spread of SARS-CoV-2 among the population could have favorited bacterial infections in the compromised immune systems patients by COVID-19 [29]. On the other hand, the use of personal protective equipment, isolation, and the increased attention to environmental sanitation to contrast the spread of the virus could probably have affected the diffusion of the secondary infection [30], and the overuse of specific antibiotics to prevent the acute symptoms, such as colistin, could have maintained and reduced, for the moment, the spread of Gram-negative bacteria in nosocomial patients (CPO+ in 2020: 37%, 2021: 35%, 2022: 28%) [31]. Still, it is possible that this trend could be inverted soon for the development of colistin resistance.

The necessity is to prevent and reduce not only the nosocomial spread of multidrug-resistant pathogens but the already worrying diffusion in the community; as observed in Figure 3a, the AMR numbers registered in this study has increased during the years, with a significant growth above all over the CPO in the community (2019: 34%, 2020: 37%; 2021: 28%, 2022: 42%). A possible explanation for this data could be the decrease in the number of hospitalizations and in access to sanitary facilities during the lockdown period [32,33] and could be the increase of antibiotics in the agro-alimentary industry, diffusing in the environmental [34]. Anyhow, it should be considered that some of the hospitalized patients were discharged in 2021–2022, becoming “Community patients” and continuing to be under health surveillance; this could contribute to the positive reverse trend. Furthermore, this statistical analysis could be biased by the intraindividual variability, gender, and age, and also by the closure of some clinical facilities during the lockdown period in favor of a private laboratory. Furthermore, self-antibiotic medication and broad-spectrum antibiotics during the COVID-19 era, such as ampicillin, erythromycin, and ciprofloxacin, were the risk factors for higher levels of Gram-positive bacteria resistance in community patients [31].

While the spread of antimicrobial-resistant bacteria in nosocomial facilities can be easily understood and, even if difficult to prevent, can be somehow contrasted, it is urgent to understand the increasing antibiotic resistance in the community, in agreement with other studies that confirmed the presence of CPO, MRSA, and VRE producers in the environment, where they can act as a reservoir of such resistance, their presence is nonetheless disturbing, as these genes can be transferred into human [34]. Moreover, many clinical and epidemiological studies indicate that antibiotic overuse can imbalance the composition of the gut microbiota, resulting in the emergence of antibiotic-resistant bacteria and the proliferation of opportunistic pathogens [35,36]. The presence of antibiotic residues in the environment can lead to contamination [37]. Therefore, the mass production and the use of antibiotics in the agro-alimentary industry need to be controlled too because alimentation is one of the factors that can modify gut microbiota and lead to the acquisition of ARB among the community. Differentiating between viral and bacterial infections can help prevent the unnecessary use of antibiotics. To this end, more accurate diagnostic tools and new molecular methods can improve the diagnosis of viral infections [38]. Additionally, the development of a new rapid detection method for bacterial infection, which still requires time-consuming procedures such as cultures, can represent an important tool to avoid unnecessary or incorrect antibiotic prescriptions [39,40,41].

## 4. Materials and Methods

### 4.1. Collecting Data

This study was conducted as a retrospective study, and the data were extrapolated by BD Phoenix (BD Company, Franklin Lakes, NJ, USA). Based on the data from a private network of Italian laboratories, Cerba HealthCare Italia, from 2019 to 2022, the aim of this study was to evaluate the incidence of multidrug-resistant pathogens among two types of patients: nosocomial patients from hospitals, clinics, and nursing homes, and community patients from drawing centers and private clinical analysis laboratories. In particular, this study evaluated the incidence of the following MDR microorganisms: *K. pneumoniae*, *E. coli*, *P. aeruginosa*, and *A. baumannii* with carbapenems resistance (CPO); *E. faecium* and *E. faecalis* with vancomycin resistance (VRE), and *S. aureus* with methicillin resistance (MRSA).

### 4.2. Collection, Identification, and Drug Susceptibility Test of Microorganisms

The cohort of microorganisms was collected from cultures of various types of specimens from bloodstream infections and non-bacteremic infections (lower respiratory tract, intra-abdominal structure, urinary tract, or other sites), according to national guidelines (Associazione Microbiologi Clinici Italiani—AMCLI). The Bruker matrix-assisted laser desorption ionization-time-of-flight mass spectrometry (MALDI-TOF MS) system (Microflex, Bruker Daltonics; Billerica, MA, USA) was used for the identification of the isolates. The BD Phoenix system (BD Diagnostics, Franklin Lakes, NJ, USA) was used for antimicrobial susceptibility testing through minimum inhibitory concentration (MIC) broth microdilution, according to European Committee on Antimicrobial Susceptibility Testing (EUCAST) criteria. In particular, for the detection of CPO isolates PHOENIX NMIC/ID503 panels were used, and for the detection of MRSA and VRE isolates PHOENIX PMIC/ID88 panels were used. 

### 4.3. Statistical Analysis

Statistical analyses were performed, and data were described by frequency. The analysis mainly involved descriptive statistics and the Chi-square test. *p*-values were based on one-tailed test results, and *p*-values < 0.05 were considered statistically significant.

## 5. Conclusions

The increases registered suggest that efforts to control and prevent the spread of antibiotic-resistant bacteria may need to be strengthened, such as through improved infection control practices and appropriate use of antibiotics. It is also important to continue monitoring these trends to track the effectiveness of interventions and to identify emerging resistance patterns.

The scientific and medical communities need to work in synergy to analyze and find the best strategies to slow down the resistance mechanism, both improving the research for better genetic isolation and characterization of genes resistance and developing new drugs or retesting known antibiotics. It is also essential to raise public awareness on the matter because the data reported in this study shows a parallel between nosocomial and community diffusion, highlighting the necessity to keep it under control in the population. It compels the world to find or develop new strategies because the antibiotics available to us are less and less effective, while the resistance amplifies due to the environmental and intrinsic characteristics of microorganisms.

## Figures and Tables

**Figure 1 antibiotics-12-00651-f001:**
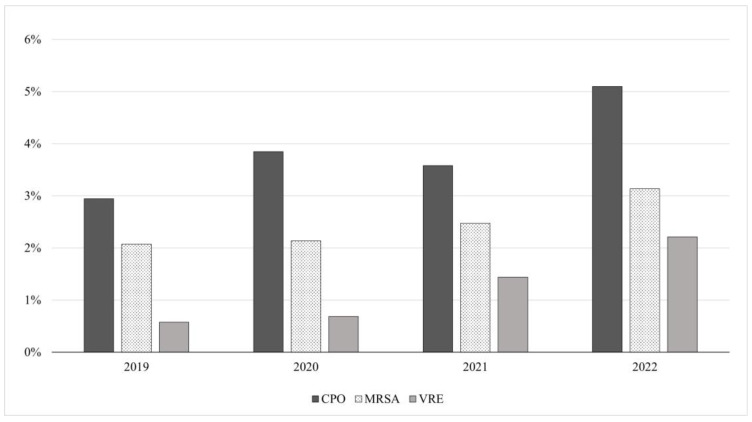
Percentages of positivity for each resistance (CPO, MRSA, and VRE), divided by year.

**Figure 2 antibiotics-12-00651-f002:**
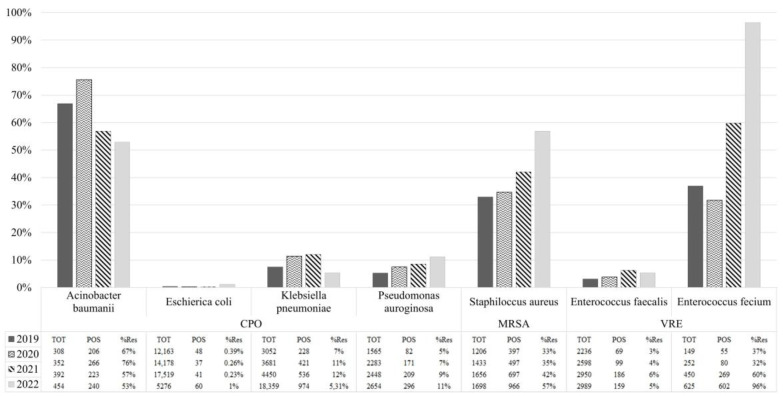
AMRs trend during the study period divided by bacterial species.

**Figure 3 antibiotics-12-00651-f003:**
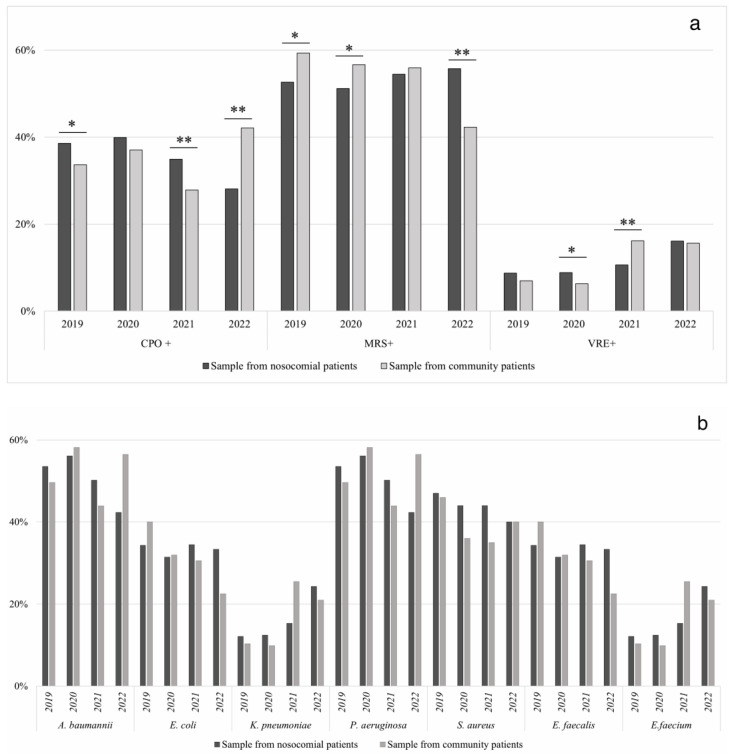
Data of resistance detection rate of samples from nosocomial patients and community patients versus CPO, MRS and VRE bacteria (panel (**a**)) and versus single pathogens (panel (**b**)) [“*” = *p* < 0.05; “**” = *p* < 0.001].

**Table 1 antibiotics-12-00651-t001:** Positive resistance percentages divided by antimicrobial substances over the four years.

	2019			2020			2021			2022		
	Total	Positive	%	Total	Positive	%	Total	Positive	%	Total	Positive	%
**CPO**	17,092	564	2.62%	20,504	895	3.43%	24,494	1009	320%	26,734	1570	4.56%
**MRSA**	2069	397	1.84%	2745	497	1.90%	3686	697	2.21%	4062	966	2.81%
**VRE**	2385	124	0.58%	2850	179	0.69%	3400	455	1.44%	3614	761	2.21%
**TOTAL**	21.546			26.099			31.580			34.410		

**Table 2 antibiotics-12-00651-t002:** Data divided between samples from nosocomial patients (N) and samples from community patients (C).

	2019				2020				2021				2022			
	Nosocomial	Community	χ^2^ Test	*p*-Value	Nosocomial	Community	χ^2^ Test	*p*-Value	Nosocomial	Community	χ^2^ Test	*p*-Value	Nosocomial	Community	χ^2^ Test	*p*-Value
**CPO+**	39%	34%	3.94	*p* < 0.05	40%	37%	1.98	*p*: ns	35%	28%	18.46	*p* < 0.001	28%	42%	95.44	*p* < 0.001
*Acinetobacter baumannii*	37%	36%	0.02	*p*: ns	30%	29%	0.20	*p*: ns	20%	24%	2.30	*p*: ns	24%	4%	121.60	*p* < 0.001
*Escherichia coli*	9%	7%	0.54	*p*: ns	7%	0%	28.09	*p* < 0.001	4%	4%	0.05	*p*: ns	6%	1%	19.18	*p* < 0.001
*Klebsiella pneumoniae*	41%	39%	0.34	*p*: ns	43%	52%	7.11	*p* < 0.05	56%	51%	2.56	*p*: ns	51%	76%	102.01	*p* < 0.001
*Pseudomonas aeruginosa*	13%	18%	2.62	*p*: ns	19%	19%	0.03	*p*: ns	20%	21%	0.28	*p*: ns	19%	19%	0.05	*p*: ns
**MRSA+**	53%	59%	6.79	*p* < 0.05	51%	57%	10.06	*p* < 0.05	54%	56%	0.73	*p*: ns	56%	42%	78.55	*p* < 0.001
*Staphilococcus aureus*	47%	46%	0.14	*p*: ns	44%	36%	7.70	*p* < 0.05	44%	35%	4.31	*p* < 0.05	40%	40%	0.00	*p*: ns
**VRE+**	9%	7%	1.55	*p*: ns	9%	6%	2.11	*p*: ns	11%	16%	20.30	*p* < 0.001	16%	16%	0.19	*p*: ns
*Enterococcus faecalis*	51%	64%	0.61	*p*: ns	53%	59%	0.19	*p*: ns	55%	34%	16.31	*p* < 0.001	17%	29%	10.98	*p* < 0.001
*Enterococcus faecium*	49%	36%	0.87	*p*: ns	47%	41%	0.25	*p*: ns	45%	66%	4.87	*p* < 0.05	83%	71%	2.07	*p*: ns

## Data Availability

Not applicable.

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
