# Peer review of "Trends in Antibiotic Resistance of Nosocomial and Community-Acquired Infections in Italy"

_antibiotics, 2023, doi:10.3390/antibiotics12040651_

Round 1
Reviewer 1 Report
The manuscript deals with a topic of great relevance, and is within the scope of the journal.
Comments:
The entire manuscript needs English correction.
Line 58: Some references do not comply with the journal's guidelines. Check throughout the manuscript.
Line 61: Genus and species must be in italics. Check throughout the manuscript.
Line 72: Check word spacing. Check throughout the manuscript.
Line 148: The figure caption should be readjusted, the term CPO in panel A does not refer to a species, but to a group of microorganisms.
The discussion is superficial. What other factors, in addition to the pandemic, could have led to the observed results?
The conclusion must be reformulated. It should resume the main points of the research, and reinforce or negate the information from other research previously presented in the introduction/discussion. New references must not be presented in the conclusion.
Author Response
Reviewer 1
Comment: The manuscript deals with a topic of great relevance, and is within the scope of the journal.
The entire manuscript needs English correction.
Response (R): Thank the Reviewer, and we increase the English form.
Comment: Line 58: Some references do not comply with the journal's guidelines
R: We modified the references, following the journal’s guidelines.
Comment: Line 61: Genus and species must be in italics. Check throughout the manuscript
R: Thanks for this comment, we corrected the typo errors.
Comment: Line 72: Check word spacing. Check throughout the manuscript.
R: Thanks for this comment, we corrected the typo errors.
Comment: Line 148: The figure caption should be readjusted, the term CPO in panel A does not refer to a species, but to a group of microorganisms.
R: We modified the Figure 3’s caption.
Comment: Line 148: The discussion is superficial. What other factors, in addition to the pandemic, could have led to the observed results?
R: We added other observations in the Discussion paragraph.
Comment: The conclusion must be reformulated. It should resume the main points of the research, and reinforce or negate the information from other research previously presented in the introduction/discussion.
New references must not be presented in the conclusion..
R: Thank you for this comment, we modified the Conclusion paragraph and removed the references, as suggested.
Reviewer 2 Report
Title: Please modify the title as Epidemiological trends in antibiotic resistance of nosocomial and community acquired infections in Italy. Considering the title (epidemiology trend) the incidence (measure) is limited to discussion and material method section only and is not derived in results. The authors need to justify the same or consider removing the epidemiological word from title and may modify it as «Trends in antibiotic resistance of nosocomial and community acquired infections in Italy».
Abstract:
Line 23: Although abbreviations are used for community and nosocomial infections; but these may be used without abbreviations or at least use two words abbreviation The use of too many abbreviations in the text reduces the readability and interest of readers. The abstract is well within word limits, hence it is easy to reduce the use of abbreviations.
Line 27: of which 11,901 resulted to be AMRs. Additionally, AMR abbreviation is in frequent use for antimicrobial resistance and not resistant. Please consider writing expanded form rather than abbreviated form for resistant.
Line 28: Rewrite the sentence as meaning is not clear.
Line 30: Clearly mentions the objectives of study viz. what authors wish to derive from the trends, the aim and design of study. Please briefly mentions the statistical findings of the study to improve the text.
Introduction:
Line 34: Again as AMR is used here for Antimicrobial resistance; refrain from using it in abstract for resistant.
Line 37: Restructure the sentence…..global priority health?
Line 42-43: Provide reference to justify the sentence …… AMR is a rising global epidemic. Improper or inappropriate…..use any single word as they are more or less synonyms.
Line 45-47: Restructure the sentence.
Line 47: Check for Reference order…[3] or [4]
Line 53: Should be written before line 47 and also restructure the sentence.
Line 59-61: Restructure the sentence. Use italics for bacteria name…..Pseudomonas
Results:
Line 117-141: Break the long sentences to short one, to improve the readability and understanding of the readers. Rewrite the sentences and also consider using decimal in place of (,) throughout the text for numerical values.
The analyses lack any correlation with age, sex and other parameters to consider it as an epidemiological study.
Discussion
Line 175-178: The findings of this study pointed the COVID-19 as the reason for increase in trend of resistance. However, the increase in trend itself seems to be very subtle for E. coli and Pseudomonas spp.. Other reasons like increase in sampling, surveillance or frequent visits to hospitals may be attributed as causes of the resistance? The authors need to cover this aspect.
Conclusion
Consider removing the references from conclusion section and report the findings of study or future insights or recommendations to authorities or organizations.
General comments
Epidemiological trends refer to the patterns or changes observed in the occurrence and distribution of a disease or health condition over time and in different populations. Epidemiological trends can be described in terms of incidence (the number of new cases of a disease in a population over a given time period), prevalence (the total number of cases of a disease in a population at a given time), and mortality (the number of deaths attributed to a disease in a population over a given time period). The study lacks correlation and analyses of results or trends with various other factors such as age group; sex; disease condition; or other demographic variables.
Author Response
Reviewer 2
Comment: Title: Please modify the title as Epidemiological trends in antibiotic resistance of nosocomial and community acquired infections in Italy. Considering the title (epidemiology trend) the incidence (measure) is limited to discussion and material method section only and is not derived in results. The authors need to justify the same or consider removing the epidemiological word from title and may modify it as «Trends in antibiotic resistance of nosocomial and community acquired infections in Italy»..
R: We thank the reviewer and considering your concern, we changed the title.
Comment: Abstract: Although abbreviations are used for community and nosocomial infections; but these may be used without abbreviations or at least use two words abbreviation The use of too many abbreviations in the text reduces the readability and interest of readers. The abstract is well within word limits, hence it is easy to reduce the use of abbreviations.
R: The abbreviations in the Abstract were explained, however, according to the 200 words limit, we reached 197 words, so it is not possible to reduce the use of them.
Comment: Abstract: Line 27: of which 11,901 resulted to be AMRs. Additionally, AMR abbreviation is in frequent use for antimicrobial resistance and not resistant. Please consider writing expanded form rather than abbreviated form for resistant.
R: Thanks for this comment, where possible, we used resistant.
Comment: Abstract: Line 28: Rewrite the sentence as meaning is not clear.
R: We thank the Reviewer, the sentence was rewritten.
Comment: Abstract: Clearly mentions the objectives of study viz. what authors wish to derive from the trends, the aim and design of study
R: According to the 200 words limit for the Abstract, we wrote the Objective, the Aim, and the Design of this study briefly, as possible.
Comment: Introduction: Line 34: Again as AMR is used here for Antimicrobial resistance; refrain from using it in abstract for resistant.
R: As answered above, where possible, we changed it to resistant.
Comment: Introduction: Line 37: Restructure the sentence…..global priority health?
R: Thanks, we restructured the sentence.
Comment: Introduction: Line 42-43: Provide reference to justify the sentence …… AMR is a rising global epidemic.
R: We added the reference in the text, justifying the sentence.
Comment: Introduction: Line 45-47: Restructure the sentence.
R: Thanks, we restructured the sentence.
Comment: Introduction: Line 47: Check for Reference order…[3] or [4]
R: We correct the order of the references.
Comment: Introduction: Line 53: Should be written before line 47 and also restructure the sentence.
R: Thanks, we restructured the sentence.
Comment: Introduction: Line 59-61: Restructure the sentence. Use italics for bacteria name…..Pseudomonas
R: Thanks, we restructured the sentence, and we corrected the typo error.
Comment: Results: Line 117-141: Break the long sentences to short one, to improve the readability and understanding of the readers. Rewrite the sentences and also consider using decimal in place of (,) throughout the text for numerical values.
R: Thanks, we restructured the sentence, and we corrected the typo error for the numerical values.
Comment: Results: The analyses lack any correlation with age, sex and other parameters to consider it as an epidemiological study.
R: Thanks to the Reviewer, we excluded the epidemiological approach.
Comment: Discussion: Line 175-178: The findings of this study pointed the COVID-19 as the reason for increase in trend of resistance. However, the increase in trend itself seems to be very subtle for E. coli and Pseudomonas spp.. Other reasons like increase in sampling, surveillance or frequent visits to hospitals may be attributed as causes of the resistance? The authors need to cover this aspect.
R: We thank the Reviewer for this insightful comment, we improved the discussion.
Comment: Conclusion: Consider removing the references from conclusion section and report the findings of study or future insights or recommendations to authorities or organizations.
R: Thank you for this comment, we removed the references, as suggested, and we added some future prospectives.
Comment: General comments: Epidemiological trends refer to the patterns or changes observed in the occurrence and distribution of a disease or health condition over time and in different populations. Epidemiological trends can be described in terms of incidence (the number of new cases of a disease in a population over a given time period), prevalence (the total number of cases of a disease in a population at a given time), and mortality (the number of deaths attributed to a disease in a population over a given time period). The study lacks correlation and analyses of results or trends with various other factors such as age group; sex; disease condition; or other demographic variables.
R: Thanks to the Reviewer, we excluded the epidemiological approach.
Reviewer 3 Report
Dear authors
The manuscript topic is in high priority and related. Howere, it is proposed to consider the following comments to improve the manuscript.
1- Reference missing in methods of identification and antibiotic resistance test.
2- The data needed to be discussed for resistance to last antibiotic remedies in other European countries.
3- Minor English language corrections are needed.
4- More suggestions for control of antibiotic resistance spread
Author Response
Reviewer 3
Comment: The manuscript topic is in high priority and related. However, it is proposed to consider the following comments to improve the manuscript.
Reference missing in methods of identification and antibiotic resistance test.
R: Thank you for this comment, and we added the method used in Material and Methods.
Comment: The data needed to be discussed for resistance to last antibiotic remedies in other European countries.
R: We thank the reviewer, we already refers the European surveillance (ECDC). Nevertheless, considering your concern, we discussed more widely this aspect in the Discussion paragraph.
Comment: Minor English language corrections are needed.
R: Thank the Reviewer, and we increase the English form.
Comment: More suggestions for control of antibiotic resistance spread.
R: Thanks, in Conclusion paragraph we included some future prospectives.
Round 2
Reviewer 1 Report
The manuscript is suitable for publication.
Reviewer 2 Report
The authors have modified the manuscript as per recommendations made. May be accepted after minor revision.